# Effects of Tetrabasic Zinc Chloride on the Diarrhea Rate, Intestinal Morphology, Immune Indices and Microflora of Weaned Piglets

**DOI:** 10.3390/ani14050737

**Published:** 2024-02-27

**Authors:** Shuyu Peng, Nan Zhang, Tuan Zhang, Yu Zhang, Shuang Dong, Huiyun Wang, Cong Xu, Chunlin Wang

**Affiliations:** 1State Key Laboratory of Animal Nutrition, College of Animal Science and Technology, China Agricultural University, Beijing 100193, China; psy991118@126.com (S.P.); zhangnan1426@163.com (N.Z.); tuan0900@163.com (T.Z.); 17753364247@163.com (Y.Z.); dongshuang19991106@163.com (S.D.); 2Changsha Xinjia Bio-Engineeriong Co., Ltd., Changsha 410300, China; why931011@126.com (H.W.); xu.cong@mail.huji.ac.il (C.X.)

**Keywords:** tetrabasic basic zinc chloride, weaned piglet, intestinal morphology, microbial community

## Abstract

**Simple Summary:**

Piglets are susceptible to a variety of diseases due to weaning stress and the shift in nutrient supply from milk to solid feed, resulting in lower disease resistance. Diarrhea is a common problem faced by the pig industry, which negatively impacts production costs. Zinc oxide (ZnO) is an effective feed additive to prevent diarrhea and promote the development of the immune system and the digestive system. However, high doses of ZnO cause pollution of soil and the environment. The use of ZnO is restricted in an increasing number of countries, so a low-dose, efficient alternative source of zinc is urgently needed by the industry.

**Abstract:**

This study was aimed to investigate the effects of different dietary zinc sources on the diarrhea rate, intestinal morphology, immune indexes and intestinal microbial composition of weaned piglets. A total of 240 weaned piglets (Duroc × Landrace × Yorkshire), at the age of 21 days, were randomly assigned to five dietary treatments for a four-week feeding trial to determine the effects of different amounts of tetrabasic zinc chloride (TBZC) supplementation on intestinal morphology, intestinal immune indices and intestinal microflora in weaned piglets, compared with the pharmacological dose of ZnO. The dietary treatments included a negative control (CON), (T1) ZnO (ZnO, 1500 mg/kg), (T2) tetrabasic zinc chloride (TBZC, 800 mg/kg), (T3) tetrabasic zinc chloride (TBZC, 1000 mg/kg), and (T4) tetrabasic zinc chloride (TBZC, 1200 mg/kg). Each treatment comprised six replicate pens, with eight pigs (four barrows and four gilts) per pen. Dietary TBZC of 1200 mg/kg improved the duodenum villus height, jejunum villus height and crypt depth of ileum, and increased the ratio of villus height to crypt depth of ileum (*p* < 0.05). The dietary supplementation of TBZC at a dosage of 1200 mg/kg has the potential to increase the levels of immunoglobulin G (IgG) and immunoglobulin A (IgA) in the duodenal mucosa. Furthermore, it shows a significant increase in the levels of immunoglobulin A (IgA) in the ileum. Compared with CON, TBZC significantly (*p* < 0.05) decreased pH values of stomach contents. It also increased the number of *Firmicutes* in intestinal contents. Compared with CON, the abundance of *Firmicutes* in jejunum contents of other treatments was significantly improved (*p* < 0.05), while the abundance of *Proteobacteria* in ileum contents of high-zinc treatments (T2 and T5) was decreased (*p* < 0.05). In conclusion, dietary TBZC of 1200 mg/kg improved the digestibility of crude protein in weaned piglets, altered the intestinal morphology of piglets, changed the intestinal microflora of piglets, reduced the diarrhea rate, and significantly improved the development of the small intestine of weaned piglets, and its regulation mechanism on intestinal tract needs further study. In summary, TBZC is likely to be an effective substitute source for the pharmacological dose of ZnO to control diarrhea in weaned piglets.

## 1. Introduction

In modern pig production, piglets are weaned at 3~4 weeks, which is much earlier than the weaning age in the normal state [1]. Due to changes in the environment, social structure, and food sources, weaned piglets often encounter a variety of stressors, which can culminate in diarrhea, intestinal inflammation and pathogenic bacteria infection and which pose a threat to the health of piglets [2]. Diarrhea of weaned piglets is the main cause of growth performance decline and even death [3]. Due to the insufficient secretion of stomach acid and digestive enzymes at weaning, nutrients cannot be fully digested, the source of lactose is reduced, and the content of lactic acid is decreased; this can lead to diarrhea caused by indigestion [4]. Additionally, the levels of maternal antibodies decline after weaning, leaving piglets with an immature immune system and making them vulnerable to infection from pathogenic bacteria. This susceptibility often results in the onset of diarrhea [5]. The gastrointestinal tract, or gut, serves multiple essential functions in piglets. It facilitates the digestion and absorption of nutrients while acting as a critical barrier against external factors and potential diseases. The structural integrity and proper functioning of the intestines significantly influence the overall health of piglets. After weaning, the intestinal microecological environment of piglets undergoes significant changes. The composition of the intestinal microbial community plays a crucial role in maintaining the overall health of the host. Therefore, promoting the healthy development of the intestine is a paramount task during the weaning stage of piglets and is integral to their overall health.

Zn is a highly essential trace element for the control of several physiological functions, including animal growth and health, immune system, and bone development, since it is a component of more than 300 different enzymes and proteins [6,7]. Zinc oxide (ZnO) is often used as one of the sources of zinc in feed; the pharmacological dose of ZnO is used as a means to control diarrhea and improve the growth performance in piglets [8]. This supplementation has been found to have a notable impact on the intestinal morphological development of piglets, bolster their immune function within the intestines, and facilitate the colonization of beneficial bacteria. Currently, zinc oxide is incorporated into feed in different forms, including coated with zinc oxide and nano-zinc oxide. However, studies have found that long-term feeding of high-dose zinc oxide can have adverse effects on the immune metabolism and intestinal function of piglets, and cause ZnO-dependent diarrhea, as well as soil and water pollution [9]. Therefore, more low-zinc piglet diets and zinc oxide substitutes have been investigated [10]. Tetrabasic zinc chloride (TBZC) is an inorganic zinc prepared by the crystallization process, which was officially approved as a feed additive in 2003. Its chemical properties are more stable and can reduce the risk of oxidation of nutrients during dietary storage. Compared with ZnO, TBZC has higher relative bioavailability and better palatability [11], and contains lower heavy metal impurities compared to other conventional inorganic zinc. It has been reported that the addition of TBZC to feed rations in place of other zinc additives such as ZnO significantly improved the growth performance of fattening pigs [12]. In previous experiments, the addition of 1000 mg/kg TBZC to the diet of piglets resulted in significant improvements in growth performance and nutrient digestibility [13]. Building on these findings, the present study aimed to investigate the effects of replacing zinc oxide with TBZC on intestinal morphology, immune status, and intestinal microbes of weaned piglets. To accomplish this, three treatment groups were designed in this experiment, receiving supplementation of 800, 1000, and 1200 mg/kg TBZC.

## 2. Materials and Methods

The experimental design and procedures used in this study were approved by the animal ethical committee of China Agricultural University (Beijing, China; No. AW11104202-1-3) according to the Chinese Guidelines for Animal Welfare. The experiment was conducted at the FengNing Swine Research Unit of China Agricultural University (Chengdejiuyun Agricultural and Livestock Co., Ltd., Chengde, China).

### 2.1. Animals and Experimental Designs

A total of 240 piglets (Duroc × Landrace × Yorkshire, BW = 6.59 ± 0.05 kg) weaned at 21 d of age were randomly divided into 5 treatments based on body weight (BW) and sex, and each treatment included 6 replicate pens and 8 piglets per pen. The 5 treatments included a control diet: (CON) control group; (T1) CON + 1500 mg/kg zinc in ZnO; (T2) CON + 800 mg/kg zinc in TBZC; (T3) CON + 1000 mg/kg zinc in TBZC; (T4) CON + 1200 mg/kg zinc in TBZC. The ZnO and TBZC used in this study were provided by Xinjia Bio-Engineeriong Co., Ltd. (Changsha, China).

The non-medicated corn-soybean basal diets were formulated to meet requirements recommended by the National Research Council (NRC, 2012) for 5 to 7 and 7 to 11 kg BW pigs and are shown in Table 1.

### 2.2. Feeding and Management

All piglets were housed in a temperature-controlled nursery (temperature 26~28 °C; humidity 55%~70%) and had ad libitum access to feed and water for 28 days. Feeders were checked and additional feed was added as needed to achieve ad libitum access at 08:30 and 15:30 daily, and their daily feed intake, feces, diarrhea situation, and feed intake were observed and recorded.

### 2.3. Diarrhea Scores

The anus of pigs was examined every day during the experimental period, and the number of pigs with diarrhea was observed and counted. The scoring system was applied to determine the rate of diarrhea as follows: 0 = hard feces; 1 = slightly soft feces; 2 = soft, partially formed feces; 3 = loose, semiliquid feces; 4 = watery, mucous-like feces. When the average score was over 2 for 2 consecutive days, pigs were identified as having diarrhea [14]. The diarrhea rate was calculated according to the following equation:Diarrhea rate (%)=the number of diarrhea pigs × diarrhea days/(the total number of pigs × experiment days) × 100

### 2.4. Sample Collections

During this experiment, feed samples of each treatment were collected at the beginning of this study and stored at −20 °C. On days 12 to 14 and 26 to 28, fresh fecal samples were collected from each pen. After mixing, the representative fresh fecal samples were baked at 65 °C for 72 h, rehumidified at natural conditions for 24 h, ground passed through a 40-mesh sieve, and stored at −20 °C until analysis.

On day 15, 6 pigs from each treatment group close to the median BW were selected and euthanized. The small intestinal tracts were taken from the abdominal cavity, and divided according to physiological features into three parts: duodenum, jejunum, and ileum [15]. A 2 cm segment from each of duodenum, jejunum and ileum were taken, and the intestinal specimens were washed with normal saline and stored in 4% paraformaldehyde for 24 h for morphological examination. Jejunum, ileum, cecum, and colon contents were collected in cryopreserved tubes and immediately frozen in liquid nitrogen and stored at −80 °C until testing for microbial composition analysis. The stomach was cut open during dissection, and the pH of the stomach contents was measured using the pH detector, then the jejunum and ileum were selected, and the contents were placed in a paper cup, and the metal probe of the detector was completely submerged in the contents for 3 s and then the values were recorded after stabilization. Then, the remaining duodenum, jejunum, and ileum segments were rinsed with cold normal saline, the intestinal segments were cut completely, the mucosal layer was carefully scraped off with slides, and the segments were immediately frozen in liquid nitrogen and stored at −80 °C for detection of IgG and IgA.

### 2.5. Chemical Analysis for Diet and Feces

Diets and feces were analyzed for crude protein (CP) and chromium using AOAC methods [16]. ATTD of nutrients is as follows:$$\mathrm{ATTD}\;\mathrm{of}\;\mathrm{nutrients}\;(\%)=\{1-({\mathrm{AIA}}_{\mathrm{diet}}\;\times \;{\mathrm{Nutrient}}_{\mathrm{feces}})/({\mathrm{AIA}}_{\mathrm{feces}}\;\times \;{\mathrm{Nutrient}}_{\mathrm{diet}})\}\;\times \;100$$

### 2.6. Histological Analysis

Intestinal tissues were stored in 4% paraformaldehyde at room temperature. The samples were cut into small pieces with a width of 2 cm. Intestinal samples were dehydrated and dealcoholized, embedded in paraffin, and sectioned. The slides were deparaffinized, rehydrated, and subjected to H&E staining. Samples were examined using an Olympus IX51 inverted phase contrast microscope.

### 2.7. Immunity of the Intestinal Tract

Firstly, IgG, IgA, and IgM in the intestinal mucosa were detected using an ELISA kit (Leibertech-nik, Beijing, China). Blank control wells were spiked with color developer A&B and termination solution, following the same steps as the other wells. A volume of 50 μL standard substance and streptavidin-HRP had been added into the standard pore. A volume of 40 μL of each sample was used for the sample wells, then 10 μL of secondary antibody and 50 μL of streptavidin-HRP were added to each well, covered with plate sealing membrane, gently shaken and mixed, and incubated at 37 °C for 60 min. The sealing plate film was removed, and the liquid was discarded. The plate was then shaken dry. Each well was filled with washing solution and allowed to stand for 30 s before discarding. This process was repeated five times. The plate was then patted dry. Subsequently, 50 μL of chromogenic agent and chromogenic agent B were added consecutively to each well and mixed. Finally, the color development was carried out at 37 °C for 10 min. A volume of 50 μL termination solution was added to each well, and the termination reaction was zero with blank conditioning. Measured the absorbance (OD) of each well sequentially at 450 nm. The measurement should be carried out within 10 min after the addition of the termination solution.

### 2.8. Microbiological Analysis of Intestinal Contents

Jejunum, ileum, cecum, and colon contents were collected in cryopreserved tubes and immediately stored at −80 °C in liquid nitrogen until testing for microbial composition analysis. Total microbial genomic DNA was extracted from the samples using the E.Z.N.A.^®^ soil DNA Kit (Omega Bio-tek, Norcross, GA, USA) according to the manufacturer’s instructions. The quality and concentration of DNA were determined by 1.0% agarose gel electrophoresis and a NanoDrop^®^ ND-2000 spectrophotometer (Thermo Scientific Inc., Waltham, MA, USA) and kept at −80 °C prior to further use. The hypervariable region V3-V4 of the bacterial 16S rRNA gene were amplified with primer pairs 338F (5’-ACTCCTACGGGAGGCAGCAG-3’) and 806R (5’-GGACTACHVGGGTWTCTAAT-3’) by an ABI GeneAmp^®^ 9700 PCR thermocycler (ABI, Los Angeles, CA, USA). The amplification cycles conditions were as follows: 95 °C pre-denaturation for 3 min, followed by 27 cycles of 95 °C denaturation for 30 s, 55 °C annealing for 30 s and 72 °C extending for 30 s, followed by 72 °C extending for 10 min. All samples were amplified in triplicate. The sequencing was performed using Illumina’s Miseq PE300 platform (Illumina, San Diego, CA, USA). The raw sequenced sequences were sequenced using fastp (https://github.com/OpenGene/fastp, version 0.19.6, accessed on 16 March 2023) software and FLASH (http://www.cbcb.umd.edu/software/flash, version 1.2.11, accessed on 16 March 2023) software for Quality control and splicing. The splicing was performed for subsequent analysis. Then, using the UPARSE software (http://drive5.com/uparse/, version 7.1, accessed on 16 March 2023), the quality control spliced sequences were subjected to operational taxonomic unit (OTU) removal of chimeras and standard clustering (with 97% confidence level) based on 97% similarity, then the representative OTUs sequences were selected for annotation. Finally, the composition of microorganisms in the gut contents was analyzed based on the standardized OTUs. The grouped samples were analyzed for differences in species composition and characteristic flora using Mothur software (http://www.mothur.org/wiki/Calculators, accessed on 16 March 2023) and the results were presented in pictures.

### 2.9. Statistical Analysis

Bioinformatic analysis of the feces microbiota was carried out using the Majorbio Cloud platform (https://cloud.majorbio.com, accessed on 16 March 2023). Based on the OTU information, the alpha diversity indices of Ace, Chao, and Shannon were calculated using Mother (http://www.mothur.org/wiki/Calculators, accessed on 16 March 2023). Bray–Curtis dissimilarity-based principal coordinates analysis (PCoA) was used to determine the similarity between microbial communities in different samples. Linear discriminant analysis (LDA) effect size (LEfSe) (http://huttenhower.sph.harvard.edu/LEfSe, accessed on 16 March 2023) was used to analyze bacterial taxa (phylum to species) that differed significantly between groups (LDA score > 2, *p* < 0.05). Differences in diarrhea rates were analyzed with the Pen (replicate) as the experimental unit. Piglets were used as experimental units to analyze crude protein digestibility, intestinal morphology, intestinal immunity and intestinal flora. Statistical analysis was performed using the unpaired t-test procedure in SAS 9.4 statistical software, ANOVA was used for analysis of variance, and Tukey’s test was used for multiple comparisons. *p* < 0.05 indicates a significant difference; 0.05 < *p* < 0.1 indicates a tendency to have a significant impact.

## 3. Results

### 3.1. The Diarrhea Rate

The effects of different zinc sources on the diarrhea rate of pigs are shown in Table 2. The addition of pharmacological zinc sources to the diets effectively reduced the diarrhea rate of weaned piglets. Both the diarrhea rates for T1, T3, and T4 were lower on days 14–28 and the whole experiment period (0.05 < *p* < 0.1) compared to CON and T2. There were no significant differences among the three groups. However, in the later stage of the experiment, the T4 group demonstrated a more pronounced anti-diarrheal effect, suggesting that a lower level of TBZC may have a similar anti-diarrhea effect as a pharmacological dose of ZnO.

### 3.2. Total Tract Digestibility of Crude Protein

The effects of different dietary zinc sources on the apparent digestibility of crude protein in weaned piglets are presented in Table 3. Compared to the control group, the T4 group showed a significant increase in crude protein digestibility (*p* < 0.01). Furthermore, the group of 1200 mg/kg of TBZC had a similar effect on crude protein digestibility as the ZnO group.

### 3.3. Intestinal Morphology

As depicted in Table 4, the dietary supplementation of 1200 mg/kg TBZC, when compared to the control group, exhibited an increase in the villus height of the duodenum, the villus height of the jejunum, and the ratio of villus height to crypt depth. However, these changes were not statistically significant. Furthermore, a significant increase was observed in the ratio of ileum villi height to crypt depth (*p* < 0.01). Conversely, no significant effects were observed in the ratio of duodenal villus height to crypt depth and the jejunum crypt depth.

### 3.4. Intestinal Mucosal Immunity

Table 5 displays the effects of various zinc sources on intestinal mucosal immunoglobulin levels in weaned piglets. In summary, the supplementation of 1200 mg/kg TBZC led to an increase in duodenal IgA content when compared to the control group, although this increase did not reach statistical significance. However, the addition of 1000 mg/kg TBZC showed a tendency to increase IgA content in the ileum.

### 3.5. PH of Gastric Contents

According to Table 6, the inclusion of ZnO (T1) resulted in a significant increase in the pH level of gastric contents compared to CON. Similarly, TBZC (T2–T4) also contributed to an increase in the pH level of gastric contents, although the effect was relatively less significant.

### 3.6. The Intestinal Microbial Composition of Piglets

As shown in Figure 1A–D, there were 76 common OTUs in the five treatment groups of jejunal contents, and 97, 3, 13, 40 and 17 OTUs were unique to the CON, T1, T2, T3 and T4 groups, respectively. A total of 49 common OTUs were unique in ileal contents from the five treatment groups, and 10, 18, 56, 12 and 14 OTUs were unique to the CON, T1, T2, T3 and T4 groups, respectively. There were 73 common OTUs in the five treatment groups for cecum contents, with 98, 9, 13, 20 and 9 OTUs unique to the CON, T1, T2, T3 and T4 groups, respectively, and 77 OTUs in the five treatment groups for colon contents, with 121, 12, 10, 16 and 6 OTUs unique to the CON, T1, T2, T3 and T4 groups.

As shown in Figure 2A–D, the principal coordinate analysis (PCoA) indicated a clear clustering of intestinal microorganisms between the control and treatment groups.

As depicted in Figure 3, the alpha diversity analysis of jejunal contents revealed significant differences (*p* < 0.05) among the treatment groups for various diversity indices. Specifically, the Ace, Chao, Coverage, Shannon, and Sobs indices showed statistically significant differences between the treatment groups. The Shannon index, in particular, was significantly higher in the CON group compared to the ZnO group. On the other hand, the Simpson index was significantly lower in the treatment groups compared to the other groups. The analysis of the ileal contents did not show any significant differences (*p* > 0.05) in the Ace, Chao, Coverage, Shannon, and Sobs indices among the five treatment groups. There were no significant differences (*p* > 0.05) in the Ace, Chao, Coverage, Shannon and Sobs indices among the cecum and colon contents groups.

Figure 4A,C,D showed that *Firmicutes*, *Proteobacteria*, *Actinobacteria*, and *Bacteroidets* are the dominant phyla in the microbial composition of the intestinal contents of weaned piglets. According to the findings depicted in Figure 5, there was a statistically significant increase (*p* < 0.05) in the abundance of *Firmicutes* in both the jejunal and cecum contents of the zinc source-added group compared to the CON group. The abundance of *Proteobacteria* showed a significant reduction in the ileal. Meanwhile, the abundance of *Proteobacteria* in the ileal and cecum contents of the high-Zn-added groups (T1 and T4) was more significantly reduced compared to the control group (*p* < 0.05).

At the species level, as shown in Figure 6A–D, *Latctobacillus_reuteri*, *Lactobacillus_johnsonil*, *Lactobacillus_salivarius* and *unclassified_g__Streptococcus* were the predominant species in the intestinal contents. Pigs in the T1 and T4 groups had a higher abundance of *Latctobacillus_reuteri* in the jejunum compared with those in the control group, but there was no difference in the ileum contents. In addition, compared with those in the CON group, the abundance of *unclassified_g__Streptococcus* in the T1, T3 and T4 groups were significantly increase in the ileal contents. In the cecum contents, *Erysipelotrichales* were higher in the CON group relative to the other treatment groups, and the relative abundance of *unclassified_o__Lactobacillales* was higher in the TBZC group. However, the relative abundance of *Negativicutes*, *Veillonellaceae* was elevated in the colonic intestinal segment in the T4 group.

## 4. Discussion

Studies have shown that the addition of pharmacological doses of zinc to weaned piglet diets can alleviate weaning stress and improve growth performance [17,18,19,20,21]. In addition to its inhibitory effect on piglet growth performance, long-term feeding of ZnO caused dependent diarrhea and soil contamination [22]. TBZC is a novel inorganic source of Zn manufactured through the reaction of high purified forms of the metal with water and hydrochloric acid [11]. It has a crystalline structure made up of covalent connections between the soluble metal ion, many hydroxyl groups, and chloride ions [23]. Mavromichalis et al. showed that weaned piglets fed 1500 mg/kg of TBZC could achieve the same growth-promoting effect as high-dose zinc oxide and were not affected by the addition of antibacterial agents, effectively improving feed conversion efficiency, the growth performance of piglets remained unaffected when TBZC was administered at levels lower than 1500 mg/kg. Furthermore, it was observed that the inclusion of 1200 mg/kg TBZC did not yield any improvements in growth performance [12]. However, the growth performance of weaned pigs decreased to the same level as the control group when TBZC was supplemented at 3000 mg/kg, which is consistent with previous report that pigs fed 3000 mg/kg Zn-deficient treatment [24,25].

Since the digestive system of weaned piglets is not fully developed, the digestibility of CP may be affected by a variety of factors; the composition of the diet, the development of the subjects, and the use of feed additives will all affect the apparent digestibility of nutrients [26]. Supplementation of 1200 mg/kg TBZC in the diet could effectively improve the digestion and absorption ability of crude protein in weaned piglets, this result is consistent with previous studies, piglets fed the TBZC diet had higher (*p* < 0.05) digestibility of crude protein and gross energy than those fed the CON diet [26]. Swine protein digestion starts in the stomach with the aid of the pepsin enzyme, which is created by the major stomach cells and transforms from an inactive form to pepsin in an acidic environment [13].

The low pH is necessary for the conversion of gastric zymogen into active enzymes. The pH of the stomach contents of the weaned piglets was higher than that of the lactation period. The increased acidity of the digestive tract affects the activity of intestinal digestive enzymes and causes proliferation of pathogenic bacteria. The feeds’ high buffering/binding capacity contributes to even higher stomach pH, it may alter gastrointestinal pH, which may have an effect on how proteins are digested and health of the gut flora [27]. The acidic properties of TBZC are comparatively lower than that of ZnO. As a result, the degree of stomach acid or acidifier consumption is also lower when using TBZC compared to ZnO. The results indicate that the addition of TBZC in the supplementation leads to a lower pH value in the stomach contents compared to the ZnO supplementation. This observation further supports the idea that TBZC provides better protection for the digestive environment in the stomach.

The small intestine is the site for digestion and absorption of nutrients in pigs. Villus height, crypt depth, and villus height to crypt depth ratio directly reflect the health of the small intestine [28]. The intestinal villi atrophy as the epithelium decreases, and the crypts deepen as cell differentiation decreases [29]. In this study, the addition of 1200 mg/kg of TBZC to the feed significantly improved the ratio of villi height to crypt depth in the ileal intestinal segment (*p* < 0.05), and there was a tendency to decrease the depth of duodenal and ileal crypt. These results indicate that TBZC has an improvement effect on intestinal morphology. However, this experiment also demonstrated that the addition of TBZC did not significantly affect the morphology of the jejunum. Previous studies have also reported that high doses of zinc did not have a significant impact on intestinal morphology at the jejunum stage of piglets [30], which aligns with the findings of this paper. Consequently, further investigation is required to fully understand the impact of TBZC on intestinal morphology.

In addition, Zinc deficiency can injure the intestinal mucosal barrier [31]. It has been shown that treating Caco2 cells with TPEN for zinc deprivation led to decreased expression of tight junction proteins, increased intestinal epithelial permeability, and impaired intestinal mucosal barrier function [32]. In this study, TBZC also tended to increase the IgA content in the duodenal and ileal mucosa, further demonstrating the positive effect of TBZC on improving the intestinal barrier. With the decrease in intestinal permeability, it also hinders the transport of harmful bacteria or toxins to a certain extent, further reducing the incidence of diarrhea and enteritis [33]. Intestinal immune indicators can reflect the health status of the intestinal tract. The results of this experiment indicate the effects of dietary supplementation with 1000 mg/kg of TBZC on the IgA content in the ileum mucosa. Although there was a noticeable increase, it did not reach statistical significance. The levels of immunoglobulin in the duodenum and jejunum mucosa were not significantly impacted. Hence, in order to gain a better understanding of the effects of TBZC on intestinal immunity, further exploration and comprehensive studies are necessary.

The colony composition and microbial diversity of the microbial community in the gut are closely related to the health of the host, which can ensure the stability and health of the intestinal function, and have a significant role in maintaining the stability of the intestinal ecosystem and reducing the stress caused by the external environment [34]. High doses of zinc can effectively inhibit the growth of harmful bacteria in the intestine [35], decrease the utilization of nutrients in the feed by microorganisms, reduce the nutrient antagonism between intestinal microorganisms and the host, thereby improving feed conversion efficiency in weaned piglets [21]. Feeding 2500 mg/kg ZnO to weaned piglets could effectively increase the microbial diversity of intestinal digestion [36]. The experimental study revealed that TBZC has the potential to enhance microbial diversity in the intestinal contents of piglets. As the microbial community becomes more diverse, there is an increase in the relative abundance of beneficial bacteria, which may contribute to improvements in piglets’ diarrhea, digestion and absorption capabilities, as well as their overall intestinal health [10]. This is consistent with the conclusion of this study that TBZC could effectively increase the abundance of beneficial bacteria and reduce the abundance of harmful bacteria in the gut of weaned pigs. However, long-term exposure to a high-zinc diet can lead to poisoning of weaned piglets, change the integrity of intestinal mucosal epithelial structure, reduce growth performance and intestinal mucosal immune function [37]. Thus, the lower dose of TBZC caused less intestinal irritation and fewer side effects in weanling piglets than ZnO.

This study demonstrated that the addition of various zinc sources had a notable impact on microbial alpha diversity in the jejunum contents. It can be inferred that zinc supplementation can modulate the diversity of microorganisms in the jejunum, potentially affecting the overall gut health and function of the subjects. However, it is important to note that there was a higher Chao index and a lower Coverage index in the CON group compared to the other treatment groups. While these differences were not statistically significant, they suggest potential variations in the microbial diversity and abundance between the CON group and the other treatment groups. TBZC was also found to significantly increase the relative abundance of *Firmicutes* in the intestine, and *Firmicutes* contribute to energy uptake and storage in pigs. Increasing the relative abundance of *Veillonellaceae* can increase the concentration of propionic acid in food, promote the production of SCFA, and improve the immune stress of piglets, *Negativicutes* can effectively promote energy uptake and storage in weaned piglets. However, there are limitations in this study to exploring the effects of TBZC on intestinal health, microflora and immune stress of weaned piglets; the effects of different doses of TBZC on the function of host-specific flora are inconsistent with previous studies, and the reproducibility is low, so further investigation is still needed.

## 5. Conclusions

Compared with ZnO, TBZC has a limited impact on the pH increase in stomach contents and the acidic environment in the gastrointestinal tract. The inclusion of 1200 mg/kg TBZC effectively enhances CP digestibility, improves piglets’ intestinal morphology, regulates the diversity of intestinal flora, and reduces the occurrence of post-weaning diarrhea. Overall, TBZC, as an alternative zinc source, exhibits favorable feeding effects in mitigating diarrhea and promoting intestinal health after weaning. In conclusion, the supplementation of TBZC in the post-weaning diet demonstrates improvements in diarrhea and intestinal morphological development, while further investigation is required to examine its effects on intestinal immune function.

## Figures and Tables

**Figure 1 animals-14-00737-f001:**
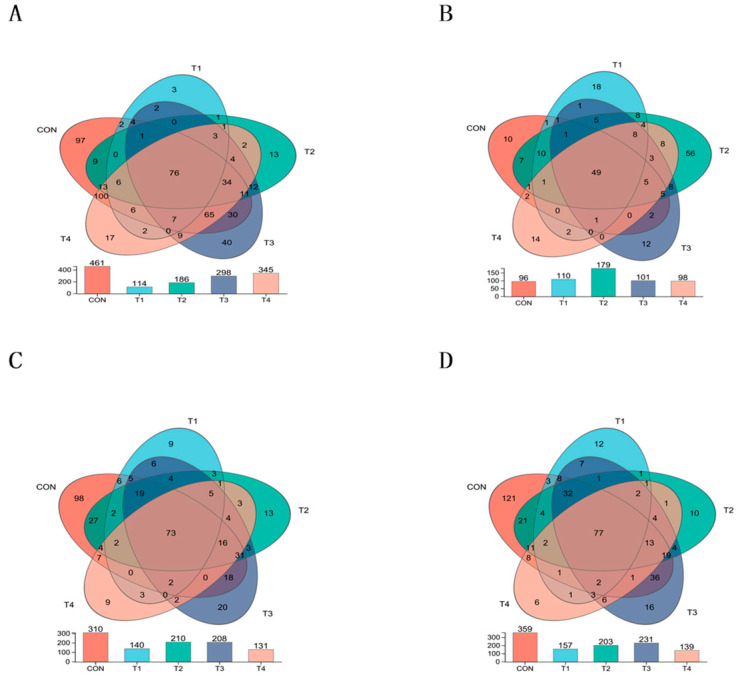
Intestinal microbial composition of Weaned piglets. Notes: (**A**–**D**) were the community composition of bacterial operation taxa of jejunum, ileum, cecum and colon contents in weaned piglets. CON, T1, T2, T3, and T4, respectively, represent the control group, 1500 mg/kg ZnO group, 800 mg/kg TBZC group, 1000 mg/kg TBZC group and 1200 mg/kg TBZC group.

**Figure 2 animals-14-00737-f002:**
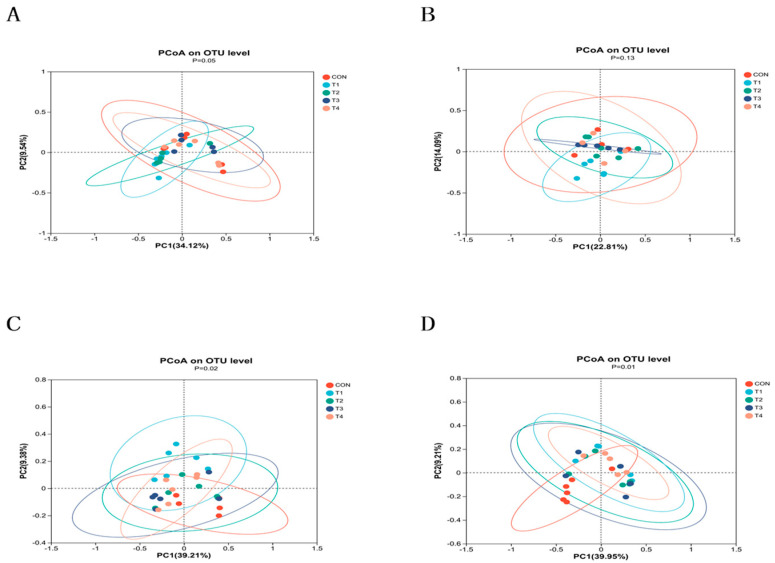
Analysis of intestinal microbial β−diversity in weaned piglets. Notes: (**A**–**D**) were β−diversity analysis of jejunum, ileum, cecum and colon contents using an unweighted version of UniFrac−based PcoA in weaned piglets. CON, T1, T2, T3, and T4, respectively, represent the control group, 1500 mg/kg ZnO group, 800 mg/kg TBZC group, 1000 mg/kg TBZC group and 1200 mg/kg TBZC group.

**Figure 3 animals-14-00737-f003:**
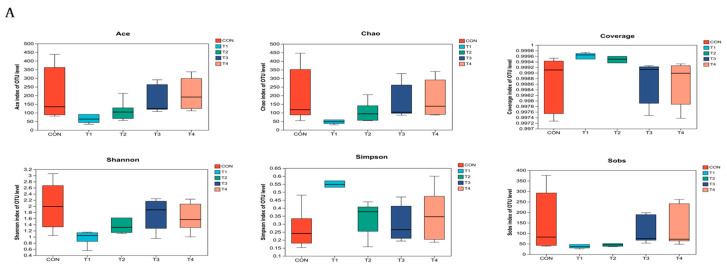
Analysis of intestinal microbial alpha diversity in weaned piglets. Notes: (**A**–**D**) were alpha diversity analysis of jejunum, ileum, cecum and colon contents based on Ace, Chao, Shannon, Coverage, Simpson and Sobs. CON, T1, T2, T3, and T4, respectively, represent the control group, 1500 mg/kg ZnO group, 800 mg/kg TBZC group, 1000 mg/kg TBZC group and 1200 mg/kg TBZC group.

**Figure 4 animals-14-00737-f004:**
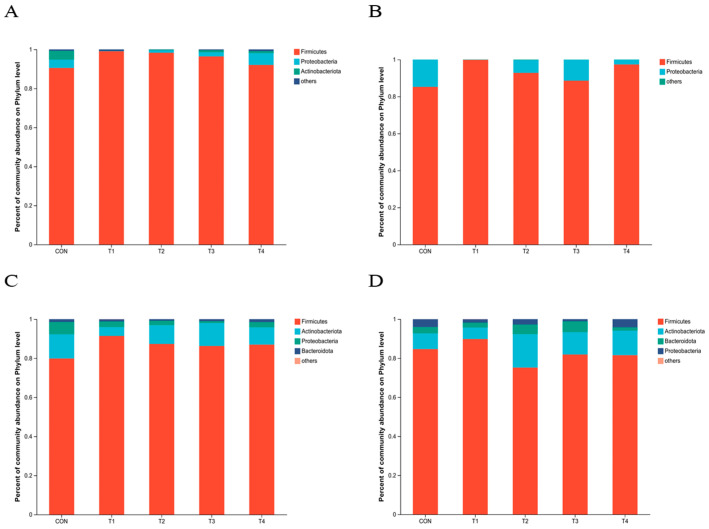
Species composition, at the phylum level, of the intestinal contents in weaned piglets. Notes: (**A**–**D**) were the species composition, at the phylum level, of the intestinal contents of jejunum, ileum, cecum and colon. CON, T1, T2, T3, and T4, respectively, represent the control group, 1500 mg/kg ZnO group, 800 mg/kg TBZC group, 1000 mg/kg TBZC group and 1200 mg/kg TBZC group.

**Figure 5 animals-14-00737-f005:**
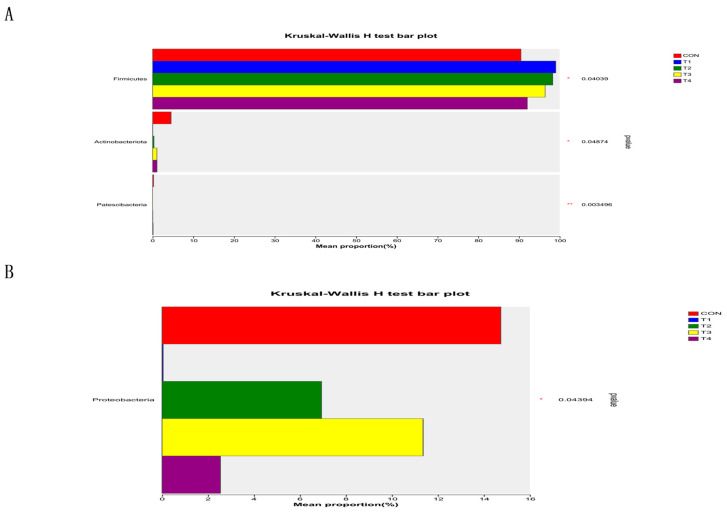
Difference of microbial content in the intestinal contents in weaned piglets at the phylum level. Notes: (**A**–**D**) were differences of microorganisms in jejunum, ileum, cecum and colon contents at the phylum level in weaned piglets. CON, T1, T2, T3, and T4, respectively, represent the control group, 1500 mg/kg ZnO group, 800 mg/kg TBZC group, 1000 mg/kg TBZC group and 1200 mg/kg TBZC group.

**Figure 6 animals-14-00737-f006:**
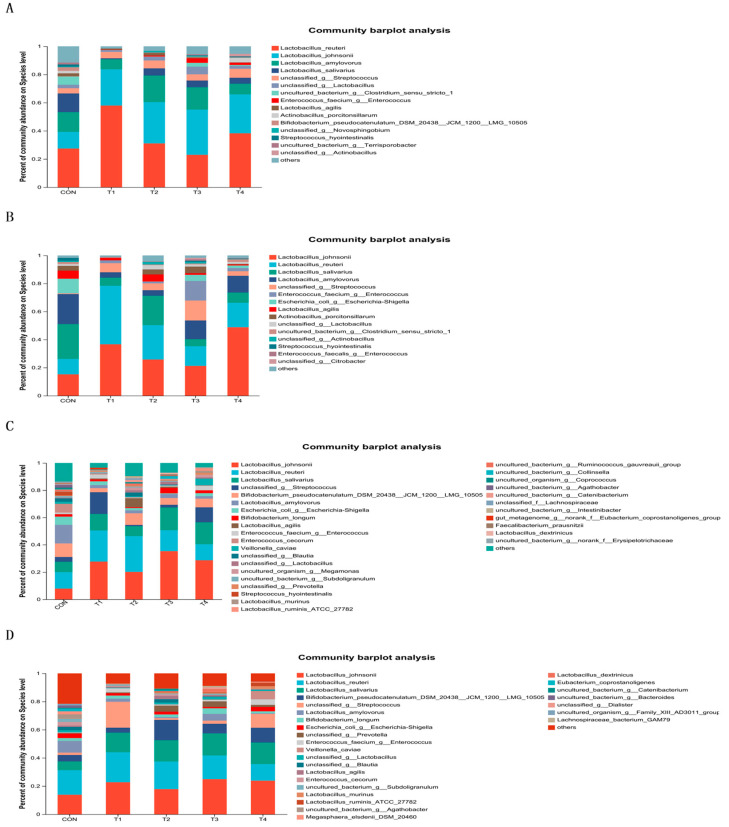
Species composition of the intestinal contents in weaned piglets. Notes: (**A**–**D**) were the composition of microbial species in the intestinal contents of jejunum, ileum, cecum and colon. CON, T1, T2, T3, and T4, respectively, represent the control group, 1500 mg/kg ZnO group, 800 mg/kg TBZC group, 1000 mg/kg TBZC group and 1200 mg/kg TBZC group.

**Table 1 animals-14-00737-t001:** The composition of the basal weaning diets (as-fed basis ^4^, %).

Items	Ingredient, % (d 1 to 14)	(d 15 to 28)
Ingredients, %		
Corn	33.12	39.00
Soybean meal, 43%	6.50	6.50
Extruded corn	30.00	30.00
Soy protein concentrate	7.00	5.00
Whey powder	8.00	5.00
Fish meal	5.00	4.00
Fermented soybean meal	5.00	5.00
Soybean oil	1.50	1.50
Dicalcium phosphate	1.30	0.42
Limestone	0.70	0.90
Salt	0.30	0.30
L-Lys HCI, 78%	0.46	0.50
DL-Methionine, 98%	0.09	0.10
L-Threonine, 98%	0.20	0.25
L-Tryptophan, 98%	0.03	0.05
Acidifier	0.30	0.30
Premix ^1^	0.50	0.50
Nutrient levels ^2^		
Digestible energy, kcal/kg	3443.29	3422.64
Crude protein, %	19.62	18.14
Calcium, %	0.80	0.73
Available phosphorus, %	0.40	0.59
SID ^3^ Lysine, %	1.50	1.42
SID Methionine, %	0.43	0.41
SID Threonine, %	0.91	0.89
SID Tryptophan, %	0.24	0.24

^1^ The premix provided the following per kilogram of diet: vitamin A, 12,000 IU; vitamin E, 30 IU; vitamin K3, 3.0 mg; vitamin B6, 6.0 mg; vitamin B12, 24 μg; vitamin B2, 10 mg; vitamin B1, 3.0 mg; nicotinic acid, 30 mg; D-pantothenic acid, 30 mg; folic acid, 2.0 mg; biotin, 0.3 mg; choline chloride, 600 mg; Cu, 10 mg; Fe, 120 mg; Zn, 120 mg; Mn, 35 mg; I, 0.3 mg; Se, 0.3 mg. ^2^ Nutrient levels are calculated according to NRC (2012). ^3^ SID means standardized ileal digestible. ^4^ ZnO and TBZC would each have to be added to replace the same amount of corn.

**Table 2 animals-14-00737-t002:** Effects of different zinc sources on the diarrhea rate (%).

Items	Treatments					SEM ^1^	*p*-Value
	CON	T1	T2	T3	T4		
0–14 d	12.11	8.97	11.85	9.47	8.48	0.67	0.08
14–28 d	6.95 ^a^	4.56 ^b^	6.53 ^a^	4.17 ^b^	2.79 ^b^	0.69	0.01
0–28 d	9.81 ^a^	7.06 ^b^	9.58 ^a^	7.14 ^b^	5.86 ^b^	0.69	<0.01

^1^ SEM, standard error of the mean (*n* = 6). ^a, b^ Different superscripts within a row mean significant difference (*p* < 0.05).

**Table 3 animals-14-00737-t003:** Effects of different zinc sources on Apparent digestibility of CP.

Items	Treatments					SEM ^1^	*p*-Value
	CON	T1	T2	T3	T4		
CP, %	68.86 ^bc^	74.35 ^ab^	62.46 ^c^	70.68 ^ab^	75.57 ^a^	0.01	<0.01

^1^ SEM, standard error of the mean (*n* = 6). ^a, b, c^ Different superscripts within a row mean significant difference (*p* < 0.05).

**Table 4 animals-14-00737-t004:** Different zinc sources on Intestinal morphology.

Item	Treatments					SEM ^1^	*p*-Value
	CON	T1	T2	T3	T4		
Duodenum							
Villi height, μm	372.07	439.10	419.28	468.43	492.65	19.15	0.33
Crypt depth, μm	444.27	377.16	358.06	363.97	330.54	14.19	0.12
Villus height/crypt depth	0.88	1.26	1.33	1.19	1.53	0.02	0.93
Jejunum							
Villi height, μm	344.98	392.78	315.42	362.77	392.62	12.94	0.27
Crypt depth, μm	291.41	306.35	308.45	345.80	313.57	13.00	0.77
Villus height/crypt depth	1.23	1.40	1.14	1.13	1.50	0.07	0.31
Ileum							
Villi height, μm	316.97	318.04	292.67	289.54	364.21	12.18	0.32
Crypt depth, μm	270.94	311.53	261.96	285.82	264.99	6.46	0.1
Villus height/Crypt depth	1.20 ^ab^	0.97 ^b^	1.18 ^ab^	1.04 ^b^	1.41 ^a^	0.05	0.01

^1^ SEM, standard error of the mean (*n* = 6). ^a, b^ Different superscripts within a row mean significant difference (*p* < 0.05).

**Table 5 animals-14-00737-t005:** Effects of different zinc sources on Intestinal mucosal immunity.

Items	Treatments					SEM ^1^	*p*-Value
	CON	T1	T2	T3	T4		
Duodenum							
IgG, μg /mg	6.64	8.78	8.24	6.07	8.57	0.44	0.22
IgA, μg /mg	11.80	15.50	13.93	10.95	15.99	1.80	0.13
IgM, μg /mg	6.09	7.73	6.26	6.14	7.62	0.74	0.39
Jejunum							
IgG, μg /mg	6.25	6.16	6.11	5.54	5.39	0.21	0.62
IgA, μg /mg	15.23	15.54	14.01	12.27	13.69	0.72	0.64
IgM, μg/mg	5.33	5.73	5.64	4.84	5.51	1.25	0.81
Ileum							
IgG, μg /mg	5.95	5.17	6.46	6.21	5.94	0.22	0.44
IgA, μg /mg	12.19	11.35	14.97	5.19	13.87	0.54	0.08
IgM, μg /mg	5.32	4.72	5.02	4.98	5.25	0.78	0.72

^1^ SEM, standard error of the mean (*n* = 6).

**Table 6 animals-14-00737-t006:** Effects of different zinc sources on pH of gastrointestinal track.

Items	Treatments					SEM ^1^	*p*-Value
	CON	T1	T2	T3	T4		
Stomach	3.92 ^a^	5.76 ^c^	4.48 ^b^	4.57 ^b^	4.88 ^b^	0.14	<0.01

^1^ SEM, standard error of the mean (*n* = 6). ^a, b, c^ Different superscripts within a row mean significant difference (*p* < 0.05).

## Data Availability

The data presented in this study are available from the corresponding author on request.

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
