# Peer review of "Effects of Tetrabasic Zinc Chloride on the Diarrhea Rate, Intestinal Morphology, Immune Indices and Microflora of Weaned Piglets"

_animals, 2024, doi:10.3390/ani14050737_

Round 1

Reviewer 1 Report

Comments and Suggestions for Authors

This paper is on the use of zinc on intestinal health weanling pigs. This is an interesting topic for both the swine industry and the scientific community. However, I believe that the paper needs to be improved before it can be considered for publication. Below are some of my comments and suggestions to improve the contents of the paper.

Simple summary: at least the authors should include what alternative products is recommended and at what dosage.

Abstract:

Lines 29: the authors stated that TBZC improved the villus height and crypt depth. Whish one of the TBZC treatments? This is applicable to all the results in the abstract.  

Line 40: which TBZC? Are the authors suggesting that the different dosage will give similar results?

Introduction:

The first paragraph end with “the transition …survival rate of piglets”. I don’t see need for this conclusion statement for this first paragraph as it introduces new topics on social hierarchy, supply, etc.

There should be some link between the paragraphs. The experiment is on digestibility, intestinal microbiome, intestinal health. I suggest the authors include these aspects in the introduction the explain what is already in the literature about ZnO and TBZC. As written, the gap in the knowledge that the experiment is supposed to close is not clear. In addition, the authors stated in line 71 that ZnSO4 significantly improved the growth…fattening pigs. So, what is the gap in the knowledge? Why is the study conducted?

Line 54: what is indigestible diarrhea?  Please, revise.

Line 106; “feeder” should start with capital letter.

Line 107: what in mental state and how was it recorded?

Materials and Methods:

There are sentence fragments throughout this this section. Examples: line 158 removed paraffine ..dry. line 162 “and then develop he color…minutes”. Lines 163-165 should also be revised to express what was done and not what should be done.

Line 196-198: suggest reorganizing the statistical section to put information together for better flow.  For examples, put all microbiome information together.

Result section.

In general, the result section needs to be totally revised. I don’t see which period the authors are referring to. What is later stage? I suggest describing the results based on the periods 0-14d, 14-28d, and 0-28 d as presented in the table.

Line 211: what is the meaning of 0.05<P<0.1?

Line 220: there was a significant increase in…T4 groups compared to what? In addition, the table 3 shows different results when we look at the mean separations. Please, describe the results accurately.

Line 226: “the addition of appropriate amount…segment” what is appropriate? Which amount are the authors referring to? Again, I suggest using reporting the exact differences. my interpretation of the table would be that T1 and T3 differ from T4 for the ration in the ileum. Again, describe the results per section. If there are no significant differences, please state it the results description. This applies to the rest of the result sections.

Line 233: “exhibit a tendency”; the authors need to define the P value range for the tendency in the statistical analysis section. In addition, stating that zinc sources exhibit a tendency to enhance IgA doesn’t convey anything concrete to the reader. There should be a comparison of the treatments. Furthermore, the authors need to show mean separation for the tendency in table 5. I suggest using x, y, or z for that.

Lines 273-275 and 288-289: these belong to the discussion.

Lines 307-317: Again, the authors need to be precise when describing the treatments differences. for example: “pigs in the T1 and T4 groups had a higher abundance …in the jejunum”, compared to which treatment groups?

Discussion:

In general, I suggest that the authors give a brief summary of the results before comparing them to previous literature and discussing their importance and their significance for the swine producers and the scientific community.

Line 323-349: these two paragraphs look like a literature review. Some of the information maybe used in the introduction to describe the problem of using ZnO, why other alternatives maybe better, what is known about TBZC, and what is missing in the literature. I suggest the authors reorganize these two paragraphs.

Line 354: “TBZC could effectively …piglet” this sentence doesn’t describe a result of the current study.

I suggest the authors give a brief summary of the results of the current study before comparing them to the reports in the literature. In addition, what was done in the previous study (Ref 28)?

Line 361-368: I don’t see the discussion the results of the current study here. What are the results of the current study?

Line 375: What is ileal recess?

The authors mentioned TBZC appropriate amount in the discussion. This term appropriate is not precise and doesn’t convey anything to the reader. In this study, the authors investigated different doses of TBZC and have different results and should discuss them accordingly.

Line 351-386: this paragraph alone discussed; digestibly, stomach pH, intestinal morphology, immunological component of the mucus layer (IgA, M, and G). I suggest splitting this paragraph in 2 or 3 small paragraphs with clear discussion and flow between them.

Line 384: in the experiment TBZC had an enhancing effect …was not significant” there is some contradiction here. If the effects of TBZC was not significant why is there an enhancing effect?

Line 396-400: “the alteration …conversion efficiency” Again this is not the description of the results of the current study. I suggest that the authors give a brief summary of the results before comparing them to previous literature and discussing the importance and the significance of their results.

Conclusion:

This conclusion is too overstated. First, which dose of TBZC? IgA, IgM, and IgG are not enough to talk about immune function of piglet and the study only investigated the intestinal mucosa. What weaning stress? There are no results on any stress parameter in the study, unless the authors are referring to diarrhea. The whole conclusion needs to be reformulate based on the results of the current study.  

Figures:

Legend fonts need to be increased for clarity and readability.

For all the figures, the authors used NOTE: A, B, C, D …

Figure caption should be more expressive and show what was done in the study and treatments used and should stand alone.

Comments on the Quality of English Language

See my comments on sentence fragments in Materials and Methods.

Author Response

List of responses

 Dear Reviewers

Thank you for your letter and for the reviewers comments concerning our manuscript entitled “Animals-2844533”. Those comments are all valuable and very helpful for revising and improving our paper, as well as important guiding significance to our researches. We have studied comments carefully and have made correction which we hope meet with approval. The main corrections in the paper and the responds to the reviewer’s comments are as flowing:

1, Response to comment: determine the specific amount of TBZC added

Response:  To determine the effects of different levels of addition on diverse indicators and ascertain the most suitable amount for addition, a thorough enumeration of these effects is necessary. For example, adding 1,200mg/kg TBZC has a significant effect on a number of indicators.

2, Response to comment: Revise the introduction and strengthen the relationship between paragraphs.

Response: The functional content descriptions of ZnO and TBZC were incorporated to elucidate the distinction between the two distinct zinc sources and explicate the knowledge gap to be addressed in the experiment.  The rationale for determining the dosage is presented, while providing a detailed description of the experimental objective.

  1. Response to comment: which period the authors are referring to. What is later stage?

Response:  The diarrhea rate was subjected to separate statistical analysis in the early and late stages of the experiment.  However, since slaughter sampling occurred on the 15th day of the experiment, there was no need for separate statistical analysis of other biochemical indicators.

4, Response to comment: what is the meaning of 0.05<P<0.1? “The addition of appropriate amount…segment” what is appropriate? The authors need to define the P value range for the tendency in the statistical analysis section, and the authors need to be precise when describing the treatments differences.

Response: Firstly, the statistical analysis section defined the P-value range of the trend.  Secondly, to clarify the specific effects of different additive contents on various biochemical indicators, the results were re-analyzed according to significance definition.  Finally, in order to enhance clarity and precision, the discussion section expression was removed from the conclusion.

5, Response to comment: These two paragraphs look like a literature review,and suggest the authors reorganize these two pargraphs, and suggest the authors  give a brief summary of the results of the current study before comparing them to the reports in the literature.

Response: To enhance the academic style, improve clarity, conciseness, and readability, the contents of the first two paragraphs in the Discussion section have been rearranged to minimize irrelevant literature discussion.  Additionally, a summary of the experimental results is included before discussing each specific project.  This approach aims to seamlessly integrate result descriptions with the subsequent discussion and analysis.

6, Response to comment: In this study, the authors investigated different doses of TBZC and have different results and should discuss them accordingly. It is recommended that gastric pH, intestinal morphology, and immune components of the mucous layer (IgA, M, G) be discussed separately in 2 or 3 small segments, with a clear discussion and flow between them.

Response: The results obtained from varying doses of TBZC are expressed more clearly and discussed in detail. Furthermore, both the pH value of the stomach and the morphology of the intestine were segmented into three sections, allowing for a comprehensive examination of TBZC's impact on the intestine from different perspectives.

7, Response to comment: Redescribe the conclusion

Response: Compared with ZnO, TBZC has a limited impact on the pH increase of stomach contents and acidic environment in the gastrointestinal tract. The inclusion of 1,200mg/kg TBZC effectively enhances CP digestibility, improves piglets' intestinal morphology, regulates the diversity of intestinal flora, and reduces the occurrence of post-weaning diarrhea. Overall, TBZC, as an alternative zinc source, exhibits favorable feeding effects in mitigating diarrhea and promoting intestinal health after weaning. In conclusion, the supplementation of TBZC in the post-weaning diet demonstrates im-provements in diarrhea and intestinal morphological development, while further in-vestigation is required to examine its effects on intestinal immune function.

Reviewer 2 Report

Comments and Suggestions for Authors

1. As the dose used in the study is high. The basis of the  dosage of ZnO and TBZC used in this study should be clarified.

2. There was a mistake in  Table 2 and 5.

3. Only the digestibility of CP was measured. Why were other nutrients digestibility not maesured?

4. As shown in the results, TBZC only impacted the ileal morphology not duodenum and jejunum. This should be discussed in discussion section.

Comments on the Quality of English Language

None.

Author Response

List of responses

 Dear Reviewers

Thank you for your letter and for the reviewers comments concerning our manuscript entitled “Animals-2844533”. Those comments are all valuable and very helpful for revising and improving our paper, as well as important guiding significance to our researches. We have studied comments carefully and have made correction which we hope meet with approval. The main corrections in the paper and the responds to the reviewer’s comments are as flowing:

1.Response to comment: As the dose used in the study is high. The basis of the  dosage of ZnO and TBZC used in this study should be clarified.

Response: In the introduction, the specific dosage of TBZC was rephrased, and the theoretical rationale behind the dosage design was emphasized.  Additionally, the abstract now includes the explicit mention of the specific dosage for each treatment.

2.Response to comment: There was a mistake in Table 2 and 5.

Response: The format of Table 2 has been revised, and an error in unit marking in Table 5 has been identified and corrected.

3.Response to comment: Only the digestibility of CP was measured. Why were other nutrients digestibility not maesured?

Response: Since it is known in other literature that high crude protein content in feed is easy to cause diarrhea of piglets, and the production cost is high, and previous studies have found that other zinc sources such as zinc oxide have a greater impact on the crude protein digestibility of piglets, this experiment also chose to explore the effect of TBZC on the protein digestibility of piglets.

4.Response to comment: As shown in the results, TBZC only impacted the ileal morphology not duodenum and jejunum. This should be discussed in discussion section.

Response: In the results section, a further analysis of intestinal morphology was conducted, specifically examining the differential effects on the ileum, jejunum, and duodenum. Additionally, the impact of TBZC on the jejunum was integrated into the discussion, and relevant literature was introduced to support the discussion.

Please see the attachment for the revised article.

Reviewer 3 Report

Comments and Suggestions for Authors

The manuscript “Effects of Tetrabasic zinc chloride on diarrhea rate, intestinal morphology, immune indices and microflora of weaned piglets” is interesting and very well written.

Weaning is the most critical period in the life of a pig, during which it has to cope with many changes in their physical and social environment. Diarrhea of weaned piglets is the main cause of growth performance decline and even death. In-feed administration of zinc oxide (ZnO) has been widely used to combat post-weaning diarrhea and growth improvement in piglets; however, in recent years, concern over use of pharmaceutical doses of ZnO has been raised as extensive use of ZnO is linked to environmental heavy metal contamination. Tetrabasic zinc chloride (TBZC) is an alternative to ZnO and can reduce the risk of oxidation of nutrients during dietary storage. Compared with ZnO, TBZC has higher relative bioavailability and better palatability. Therefore, the manuscript investigated the effects of replacing zinc oxide with TBZC on intestinal morphology, immune status, and intestinal microorganisms in weaned piglets.

The topic of the manuscript is interesting and very relevant. The manuscript has correctly pointed out the research gap and addressed it. The authors have prepared an original and well-organized study. However, below are few points to consider for the improvement of the manuscript

1. The introduction section is well organized.

2. The authors supplemented TBZC at the dose of 800, 1000 and 1,200 mg/kg (T2, T3 & T4). What was the basis of selecting these doses of TBZC? Based on any previous literature or based on any pilot study? The authors are requested to clarify.

3. The authors mentioned that “The supplementation of zinc sources in the feed exhibited a tendency to enhance the IgA content in the ileum”. From table 5, no significant difference among the groups was observed. The authors are requested to rephrase the sentence.

4. The authors didn’t mention the growth performance of the piglets at the end of the experimental period. Body weight, average daily gain and feed to gain ratio are vital parameters of animal performance in pig industry. Therefore, readers may be interested to know the effect of TBZC supplementation on growth performance of the animals.

5. The results and the discussion are presented clearly, however, I suggest the authors to strengthen the discussion with few more related literatureMoreover, the authors are requested to mention the limitations of the study and any future experiments.

Comments on the Quality of English Language

Minor editing is required. 

Author Response

List of responses

 Dear Reviewers:

Thank you for your letter and for the reviewers comments concerning our manuscript entitled “Animals-2844533”. Those comments are all valuable and very helpful for revising and improving our paper, as well as important guiding significance to our researches. We have studied comments carefully and have made correction which we hope meet with approval. The main corrections in the paper and the responds to the reviewer’s comments are as flowing:

1.Response to comment: The authors supplemented TBZC at the dose of 800, 1000 and 1,200 mg/kg (T2, T3 & T4). What was the basis of selecting these doses of TBZC? Based on any previous literature or based on any pilot study? The authors are requested to clarify.

Response: In the introduction section, the specific dosage of TBZC is reiterated, with an emphasis on the theoretical basis for dosage design. Additionally, relevant literature is cited to provide further theoretical support.

2.Response to comment: The authors mentioned that “The supplementation of zinc sources in the feed exhibited a tendency to enhance the IgA content in the ileum”.  From table 5, no significant difference among the groups was observed.  The authors are requested to rephrase the sentence.

Response: In the results and discussion section, the analysis of intestinal mucosal IgA, IgM, and IgG content was revised. Additionally, significance classification was added to the statistical analysis, and a more accurate statistical analysis of their content was conducted.

3.Response to comment: The authors didn’t mention the growth performance of the piglets at the end of the experimental period. Body weight, average daily gain and feed to gain ratio are vital parameters of animal performance in pig industry. Therefore, readers may be interested to know the effect of TBZC supplementation on growth performance of the animals.

Response: Regrettably, no statistical analysis was performed on the growth performance of piglets in this experiment, hence the absence of relevant data regarding the impact of TBZC on growth performance. I sincerely apologize for this limitation.

4.Response to comment: The results and the discussion are presented clearly, however, I suggest the authors to strengthen the discussion with few more related literature. Moreover, the authors are requested to mention the limitations of the study and any future experiments.

Response: To enhance the academic style, improve clarity, conciseness, and readability, the contents of the first two paragraphs in the Discussion section have been rearranged to minimize irrelevant literature discussion.  Additionally, a summary of the experimental results is included before discussing each specific project.  This approach aims to seamlessly integrate result descriptions with the subsequent discussion and analysis. The results obtained from varying doses of TBZC are expressed more clearly and discussed in detail. Furthermore, both the pH value of the stomach and the morphology of the intestine were segmented into three sections, allowing for a comprehensive examination of TBZC's impact on the intestine from different perspectives. Additionally, when combined with the analysis results, it becomes evident that TBZC has limited effects on intestinal morphology and no significant effects on the morphology of the jejunum stage. Moreover, TBZC does not exhibit a notable impact on intestinal mucosal immunoglobulin content. These findings indicate limitations in exploring the effects of TBZC on piglet intestinal health, necessitating further studies to elucidate this aspect.

Please see the attachment for the revised article.

Round 2

Reviewer 3 Report

Comments and Suggestions for Authors

The authors have addressed the concerns raised . I am satisfied with the responses. The authors mentioned that due to lack of relevant data on growth performance, effect of TBZC on growth parameters was not included in the manuscript. Growth parameters are very important in these kind of studies. In future, the authors are requested to include growth parameters. Overall, the manuscript has been improved significantly. 

Comments on the Quality of English Language

Minor editing required.